# Defining human mesenchymal and epithelial heterogeneity in response to oral inflammatory disease

Ana J Caetano[1†], Val Yianni[1†], Ana Volponi[1], Veronica Booth[2], Eleanor M D'Agostino[3], Paul Sharpe[1]*

[1]Centre for Craniofacial and Regenerative Biology, Faculty of Dentistry, Oral & Craniofacial Sciences, King's College London, London, United Kingdom; [2]Department of Periodontology, Faculty of Dentistry, Oral & Craniofacial Sciences, King's College London, London, United Kingdom; [3]Unilever R&D, Colworth Science Park, Sharnbrook, Bedfordshire, Bedford, United Kingdom

**Abstract** Human oral soft tissues provide the first barrier of defence against chronic inflammatory disease and hold a remarkable scarless wounding phenotype. Tissue homeostasis requires coordinated actions of epithelial, mesenchymal, and immune cells. However, the extent of heterogeneity within the human oral mucosa and how tissue cell types are affected during the course of disease progression is unknown. Using single-cell transcriptome profiling we reveal a striking remodelling of the epithelial and mesenchymal niches with a decrease in functional populations that are linked to the aetiology of the disease. Analysis of ligand–receptor interaction pairs identify potential intercellular hubs driving the inflammatory component of the disease. Our work establishes a reference map of the human oral mucosa in health and disease, and a framework for the development of new therapeutic strategies.

*For correspondence:
paul.sharpe@kcl.ac.uk

†These authors contributed equally to this work

## Introduction

The oral mucosa is one of the most rapidly dividing tissues in the body and provides the first line of defence against the development of oral disease. Gingiva is the oral mucosa that surrounds the cervical portion of the teeth, and consists of a keratinised stratified squamous epithelium and an underlying connective tissue containing multiple cell types that collectively orchestrate tissue homeostasis during health and in response to mechanical and microbial challenges (*Lindhe et al., 2008*; *Cekici et al., 2014*). Periodontal disease is a chronic inflammatory condition associated with a dysbiosis of the commensal oral microbiota and host immune defences causing irreversible destruction of the soft and hard supporting tissues of the teeth (*Pihlstrom et al., 2005*; *Lindhe et al., 2008*). Gingivitis is a mild and reversible inflammation of the gingiva that does not permanently compromise the integrity of the tissues supporting the teeth. Chronic periodontitis occurs when untreated gingivitis progresses to the loss of the gingiva, bone, and ligament (*Lamont and Hajishengallis, 2015*; *Pihlstrom et al., 2005*; *Lindhe et al., 2008*). Regenerating lost tissues remains the fundamental therapeutic goal and to achieve this it is necessary to understand the mechanisms and pathways controlling disease progression while identifying novel candidates for intervention.

Most studies on the pathogenesis of periodontal disease have largely focused on characterising the microbial biofilm and host immune response (*Hajishengallis, 2014*; *Yucel-Lindberg and Båge, 2013*). However, it is recognised that tissue resident cells play an instrumental role in innate immunity, immune regulation, and epithelial barrier maintenance (*Krausgruber et al., 2020*). Additionally, individual molecules known to play important roles in disease pathogenesis and the cell types they originate from remain ill-defined (*Yucel-Lindberg and Båge, 2013*).

Here, we set out to unbiasedly profile human gingiva, including epithelial, mesenchymal and immune compartments using single-cell RNA sequencing. To better characterise the dynamics of disease progression we used samples isolated from healthy and diseased patients. Our single-cell analysis identified differences in the composition of cellular sub populations residing within the gingival tissues and changes in the transcriptional fingerprint between healthy and diseased patient samples. We showed that these changes correlate with progressive diseased states.

Despite the growing recognition that mesenchymal (stromal) cells maintain epithelial barrier integrity and immune homeostasis in several organs (*Kabiri et al., 2014*; *Nowarski et al., 2017*; *Bernardo and Fibbe, 2013*), the identity of gingiva-specific mesenchymal subtypes and the molecular attributes that regulate niche maintenance or disease remodelling have not been described. Significantly, we identified specific changes in mesenchymal cell populations indicative of playing a role in disease progression.

Intercellular network reconstruction in healthy and diseased states revealed loss of cell communication and increased immune interactions between the identified cell types. We provide novel insights into altered communication patterns between epithelial and mesenchymal cells caused by the inflammatory response.

Taken together, our data characterise the cellular landscape and intercellular interactions of the human gingiva, which enables the discovery of previously unreported cell populations contributing to oral chronic disease. Understanding the crucial roles of individual cell states during disease progression will contribute to the development of targeted cell-based approaches to promote regeneration or reduce inflammation-associated tissue dysfunction.

## Results

### Generation of the gingival transcriptional landscape in health and periodontitis

Similar to other tissues in the gastrointestinal tract, the oral mucosa is a good model for studying a rapidly renewing tissue. To provide an in-depth analysis of cellular architecture, cell heterogeneity, and understand gingival cell dynamics when transitioning from health to disease, we transcriptionally profiled single cells derived from patients. We obtained freshly resected human gingival tissue and isolated live cells (*Figure 1—figure supplement 1*) to be sequenced on the 10x Genomics Chromium platform for single-cell RNA-seq (scRNA-seq) (*Figure 1A*). A total of 12,411 cells were captured across four patient biopsies, allowing us to perform an in-depth analysis of single-cell transcriptomics. In order to ascertain the extent of likely human variation between datasets we first compared data from two healthy patients. Cells from these healthy patients were remarkably similar (*Figure 1—figure supplement 2*) and we observed a strong linear relationship in gene signatures between the two patient samples (*Figure 1—figure supplement 2*). Having established a high concordance of datasets obtained from two biopsies of healthy gingiva and to amplify the power of the study, these were merged and handled together for the subsequent analysis.

Carrying out a comparative bioinformatic analysis of samples obtained from healthy and periodontitis patients revealed a diversity in epithelial, stromal, endothelial, and immune cells. A total of 16 distinct transcriptomic signatures were detected that corresponded to cell types or sub-populations of identifiable cell states. These were visualised using UMAP (*Figure 1B*).

In the epithelial compartment, we identified three subsets (clusters 1, 8, and 12), potentially corresponding to distinct differentiation stages. Cluster 1 shows a basal cell state with expression of *HOPX, IGFBP5,* and *LAMB3*; and cluster 8 a more mature cell state with expression of *KRT1, KRT8, LAT* (Linker for Activation of T cells) and *PTGER* both required for TCR (T-cell antigen receptor) signalling (*Figure 1B,C*; *Figure 1—figure supplement 3*).

Proliferating basal cells were identified in cluster 12 by expression of canonical marker genes of proliferating cells such as *MKI67* and *TOP2A* (*Whitfield et al., 2006*; *Figure 1B,C*; *Figure 1—figure supplement 2*). We also identify a mesenchymal (stromal-fibroblast) (cluster 6) based on collagen expression; one perivascular (cluster 10) by high expression of *PDGFRB* and *RGS5* (*Figure 1B,C*; *Figure 1—figure supplement 3*); two endothelial (clusters 9 and 15) in which cluster 9 specifically expresses *CLDN5* and *EMCN* and cluster 15 shows high expression of genes involved in the

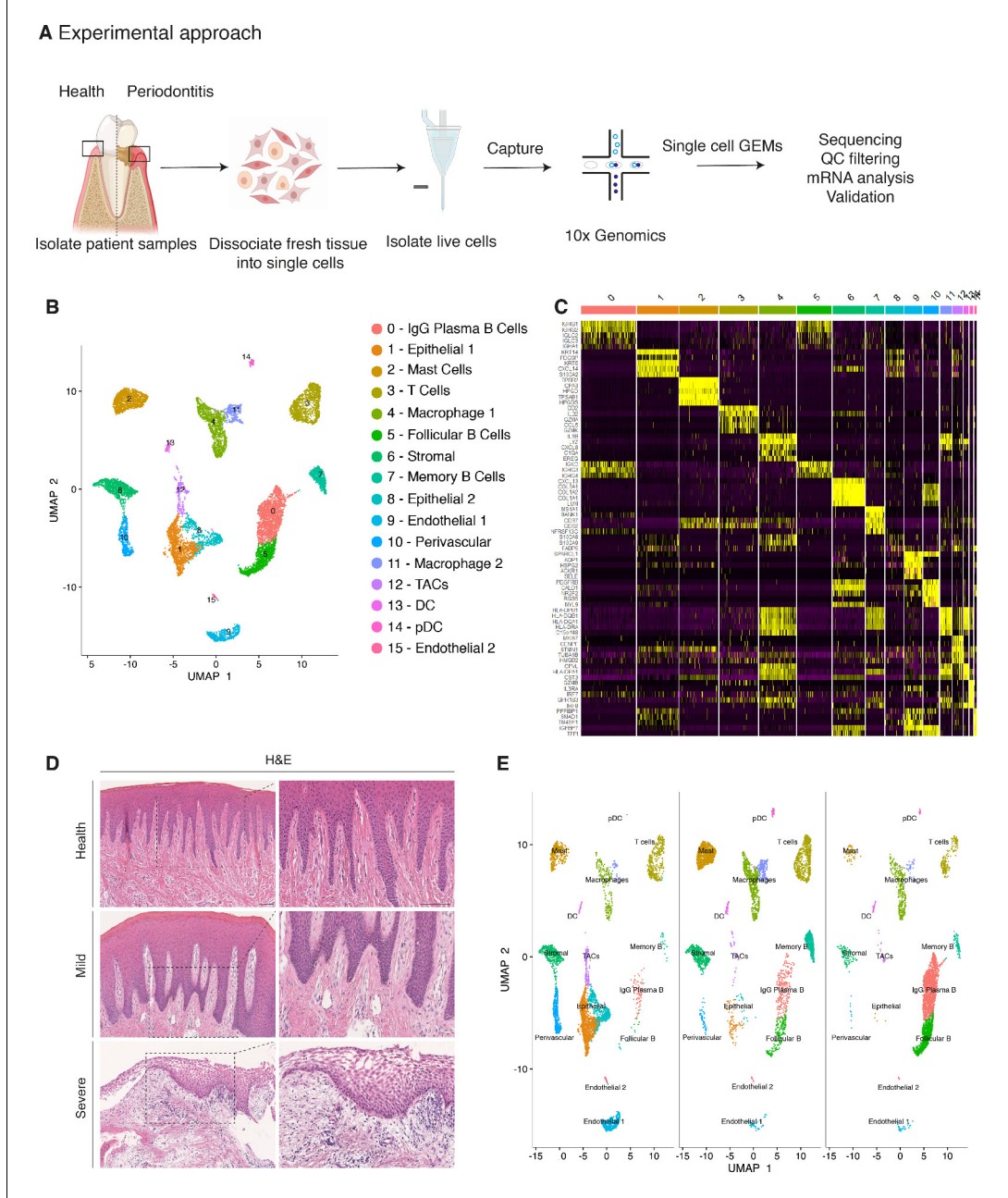

**Figure 1.** Single-cell Atlas of Gingiva Biopsies from Healthy Individuals and Periodontitis Patients. (**A**) Overview of the experimental workflow. All samples were processed immediately after clinical surgery. (**B**) scRNA-seq data obtained from healthy and periodontitis cells (n = 12,411) from four donors illustrated by UMAP coloured by cell-type annotation (nHealthy = 4639, nMild = 4401, nSevere = 3367). (**C**) Heatmap of the mean expression of the most differentially expressed marker genes for each cluster identified. (**D**) Haematoxylin and eosin staining of gingival sections from healthy, mild, and severe patient samples showing increasing changes in tissue architecture with loss of epithelial rete ridges definition and infiltration of leukocytes. (**E**) Changes in tissue composition in periodontitis showing UMAP of progressive diseased states from healthy, mild, and severely diseased donors. The online version of this article includes the following figure supplement(s) for figure 1:

**Figure supplement 1.** Flow Cytometry Gating Strategies on Human Gingival Cells.

**Figure supplement 2.** Single-cell profiling of healthy human gingiva datasets using 10x Chromium.

**Figure supplement 3.** Single-cell profiling of healthy and disease human gingiva using 10x Chromium.

regulation of angiogenesis such as *KDR*, *TIE1*, and *SOX18* (*Jones et al., 2001*; *François et al., 2008*; *Figure 1B,C*; *Figure 1—figure supplement 3*).

We identified immune clusters of the myeloid (macrophages and dendritic cells) and lymphoid (T and B cells) lineages. B cells are shown in three distinct populations (clusters 0, 5, and 7) with clusters 0 and 5 expressing *MZB1*, *DERL3*, and *IGHG4* characteristic of follicular and IgG plasma B cells respectively, and cluster 7 expressing *MS4A1* and *CD37* corresponding to memory B cells (*Akkaya et al., 2020*; *James et al., 2020*; *Figure 1B,C*; *Figure 1—figure supplement 3*). T cells are shown in cluster 3 identified by expression of canonical TRM marker *CXCR6*. Dendritic cells of myeloid origin with high expression of *CLEC9A* and *IRF8* are found in clusters 13 and 14 (*Eisenbarth, 2019*; *James et al., 2020*), and mast cells are indicated in cluster 2 expressing *TPSB2* and *TPSB1* (*Abraham and John, 2010*; *Figure 1B,C*; *Figure 1—figure supplement 3*). Macrophages are found in two populations (clusters 4 and 11) sharing high expression of *LYZ* and *AIF1* (*Chakarov et al., 2019*; *Figure 1B,C*; *Figure 1—figure supplement 3*). In most cases, further subclusters could not be distinguished by an additional clustering, with the exception of epithelial and stromal clusters.

Together, these data provide the first detailed molecular insight into gingival cell populations supported by known and novel markers.

## Transcriptional comparison of healthy and periodontitis reveals progressive diseased states

During disease progression there is a distinct signature of clinical phenotypes including redness, swelling, bleeding, destruction of periodontal ligament and bone and gingival recession (*Kinane, 2001*). These clinical manifestations are due to the dysregulation of a number of cell types which include epithelial, stromal, immune, and the associate cross-talk between them (*Pihlstrom et al., 2005*).

Histologically, the diseased samples showed different levels of severity. Therefore, in our analysis we staged the samples as healthy, mild, and severe. (*Figure 1D*). In the mildly affected sample, we observed an intact keratinised squamous epithelial layer, minor losses of collagen, and rete-ridge definition. In contrast, in the severe state we detected a dense infiltrate of lymphocytes, breakdown of the epithelial barrier and clear reduction of collagen content (*Figure 1D*).

To investigate the transitions between health and mild to severe periodontitis, we determined the contribution of cells sampled from each condition to the main cell classes, and investigated whether their respective subpopulations were maintained, amplified or depleted across the conditions.

At a transcriptomic level, the cellular landscape is dominated by a corresponding shift in cellular proportions (*Figure 1E*). In health, we observed low numbers of follicular and plasma B cells and a progressive increase from mild to severe (*Figure 1E*; *Figure 1—figure supplement 3*). The minimal presence of B cells in healthy gingiva was also reported by others (*Dutzan et al., 2016*; *Mahanonda et al., 2016*; *Artese et al., 2011*). Memory B cells show a distinctive increase in the mild sample with a subsequent decrease in the severe sample (*Figure 1E*).

Similarly, there was a surge in T cells in mild disease followed by a decrease in severe. While there has been some characterisation of immune cell subsets in health and periodontitis (*Dutzan et al., 2016*), the timing of their involvement is still unclear. Our study addresses this to some extent by showing that these populations may be abundant in the mild stage and then gradually decrease as disease progresses. T-cell senescence as a result of persistent immune activation in chronic diseases has been previously reported (*Effros and Pawelec, 1997*; *Vallejo et al., 2004*). A decrease in the severe stage might suggest that the persistent immune activation characteristic of chronic inflammation may lead to T-cell senescence, and consequently to the inability to reduce local inflammatory responses contributing to disease persistence. Additionally, we also identified a dynamic shift in the two macrophage populations with an expansion in the mild stage consistent with their function in tissue clearing and a subsequent reduction at the severe stage (*Figure 1E*). There is no clear difference in the dendritic cell compartment during disease progression. Mast cells also show a significant enrichment in mild and a decrease in the severe state. These results deliver the first unbiased immune characterisation of the gingiva across disease states (*Figure 1E*).

In addition to infiltrating immune cells driving the inflammatory process, mesenchymal and epithelial gingival cells in the gingiva are also affected during the progression and persistence of the

disease (*Yucel-Lindberg and Båge, 2013*). We observed a progressive depletion of both mesenchymal and epithelial cell populations (*Figure 1E*), in line with the patient matched immunohistochemical studies.

Together these results provide with the first comprehensive platform to compare dynamic changes of gingival cell populations during disease development.

## Cellular and molecular map of the stromal gingival compartment in health and disease identifies subpopulations with potential roles in disease progression

Tissue mesenchymal cells play essential roles in epithelial homeostasis, matrix remodelling, immunity, and inflammation (*Kinchen et al., 2018*; *Nowarski et al., 2017*). Their function in the regulation of acute and chronic inflammation in peripheral organs is now well established (*Fiocchi et al., 2006*; *Kinchen et al., 2018*; *Croft et al., 2019*). Despite the growing recognition that the mesenchyme acts as a critical regulator in disease persistence by producing cytokines, chemokines, proteolytic enzymes, and prostaglandins (*Yucel-Lindberg and Båge, 2013*), the identity of gingiva-specific mesenchymal subtypes and the molecular attributes that regulate niche maintenance in disease have not been described. To better visualise the difference in cellular heterogeneity of gingival stromal cells in health and disease, we performed re-clustering analysis of collagen expressing cells to identify any possible sub-clusters with a distinct transcriptional signature.

These data revealed five fibroblast-like populations, one pericyte and one myofibroblast (*Figure 2A*; *Figure 2—figure supplement 1*). Myofibroblasts were identified by expression of *ACTA2* and by gene ontology (GO) terms such as 'muscle contraction' and 'smooth muscle contraction'. Pericytes were identified by *PDGFRB* and *MCAM* expression and GO terms such as 'regulation of angiogenesis' (*Figure 2C,D*). S0, S2, and S4 fibroblast-like subpopulations showed enrichment for genes annotated with 'extracellular matrix'-related GO terms. Interestingly, one of the fibroblast-like populations (S0) GO enrichment included 'upregulation of fibroblast proliferation' with marked expression of *PDGFRA*, *WNT5A,* and *IGF1*. It also shows upregulation of *POSTN* which is essential for tissue repair (*Kühn et al., 2007*). Another fibroblast-like population (S2) showed enrichment for genes involved in the negative regulation of Wnt signalling (*GREM1*, *SFRP1*, *APCDD1,* and *DKK3*); S4 showed expression of *OSR2*, *FGFR1*, *SOX4,* and *TBX3* known to be involved in skeletal development. Additionally, S4 also differed in the expression of a specific form of collagen, collagen IV, which is known to be a key component of the epithelial basement membrane and might suggest a role in epithelial barrier membrane as previously described (*Kinchen et al., 2018*). Finally, S5 and S6 show a potential role in immune regulation with enrichment for 'cytokine-mediated signalling pathway', 'IFN-γ signalling' and 'T-cell activation' (*Figure 2—figure supplement 1*; *Source data 2*). Highly ranked S5 markers included *ILR1*, IFNγR1, and a member of the TNF-receptor superfamily – *TNFRS11B* (osteoprotegerin) which is a negative regulator of bone resorption and thus a key regulator of osteoclast activity (*Zaidi, 2007*).

To uncover the role of the newly identified mesenchymal subsets in periodontitis, we investigated changes in their contribution across diseased states. Most significantly, we identified a marked decreased in the myofibroblast (S1) and pericyte (S3) subpopulations in the mild stage, while the other fibroblast-like cells appeared unchanged with the exception of S6 (*Figure 2B*). This suggests loss of S1 and S3 cells was the most pronounced change from healthy tissue to mild disease. We further explored the nature of the pro-inflammatory cluster S6, and it included the expression of the major histocompatibility complex (MHC) class II invariant chain (*CD74*) and *AREG* (amphiregulin). Amphiregulin is a reparative cytokine previously described with a role in gingival immune homeostasis (*Krishnan et al., 2018*). These results identified the potential expansion of a novel stromal population enriched for pro-inflammatory genes in periodontitis.

Next, we investigated whether we could detect these changes using immunofluorescence analysis in gingival tissue samples. We confirmed a decrease in collagen VI levels suggesting overwhelming changes in the ECM composition and deposition (*Figure 2E*). We also assessed the myofibroblast population by looking at expression of *ACTA2* (*Figure 2E*).

Understanding the pathways underlying stromal differentiation will be essential to understand tissue homeostasis in chronic diseases. Given the lack of markers to reconstruct a cellular trajectory and the knowledge that the number of expressed genes per cells is a hallmark of developmental potential (*Teschendorff and Enver, 2017*; *Han et al., 2020*), we used transcriptional diversity to

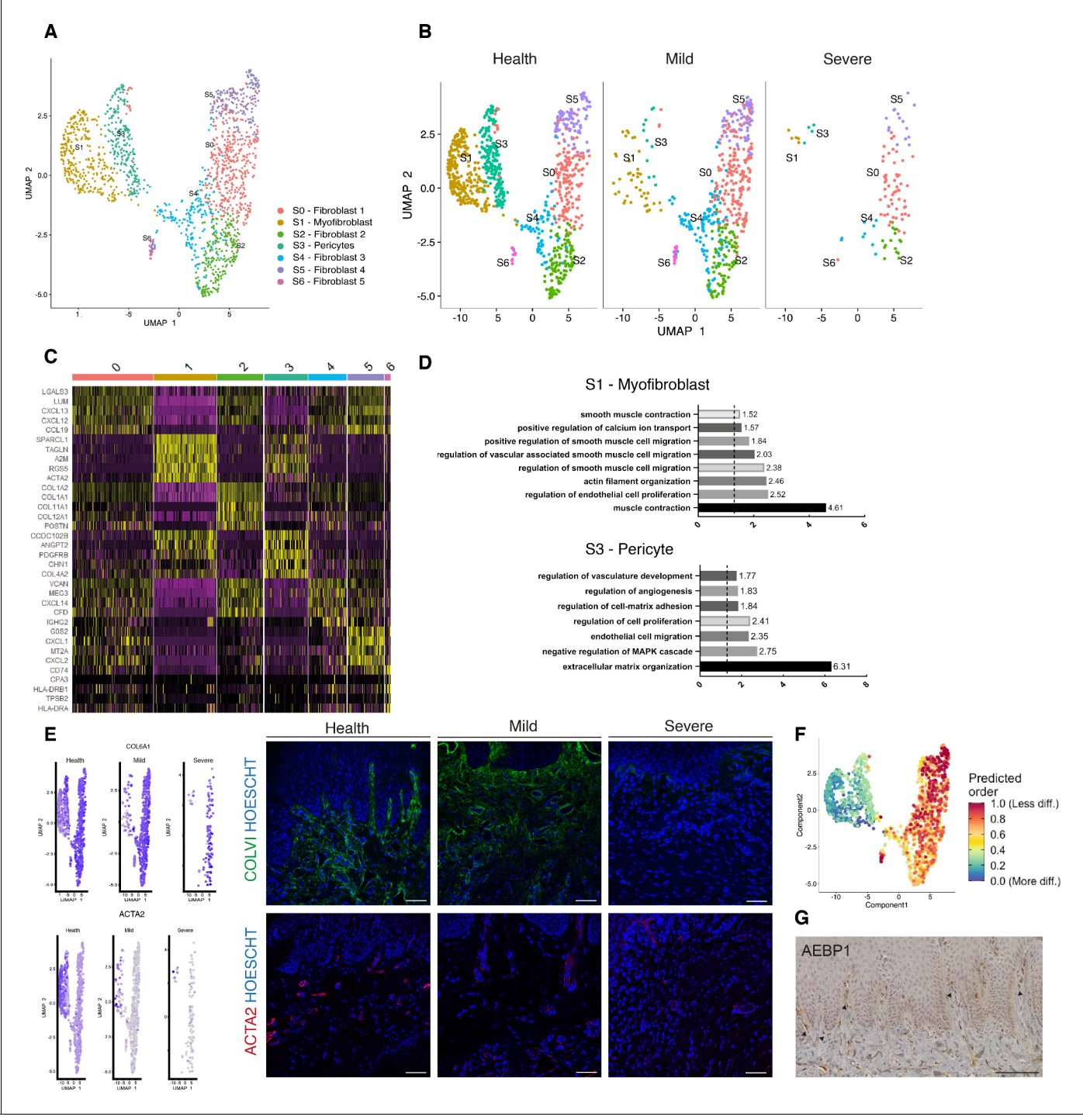

**Figure 2.** Cellular and molecular map of the stromal gingival compartment in health and disease identifies subpopulations with potential role in disease progression. (A) UMAP plot of gingival stromal cells. Single cells coloured by cluster annotation. (B) UMAP plot of stromal cells during disease progression. (C) Heatmap showing subset-specific markers. (D) GO enrichment terms for S1 (myofibroblast) and S3 (pericyte). -log adjusted p-value shown (dotted line corresponds to FDR = 0.05). (E) Immunofluorescence staining showing COLVI, ACTA2 expression throughout disease progression. Scale bars, 100 μm. n = 3 patient samples/condition. Feature plots showing COLVI and ACTA2 expression across clusters and conditions. (F) UMAP annotated with CytoTRACE analysis to predict stromal stem populations. Transcriptional diversity is used here to predict maturation states. (G) Immunohistochemistry staining showing AEBP1+ cells tissue distribution. Scale bar, 100 μm. n = 6 patient samples.

The online version of this article includes the following figure supplement(s) for figure 2:

**Figure supplement 1.** Re-clustering of human stromal gingival cells in health and disease, Related to *Figure 2*.

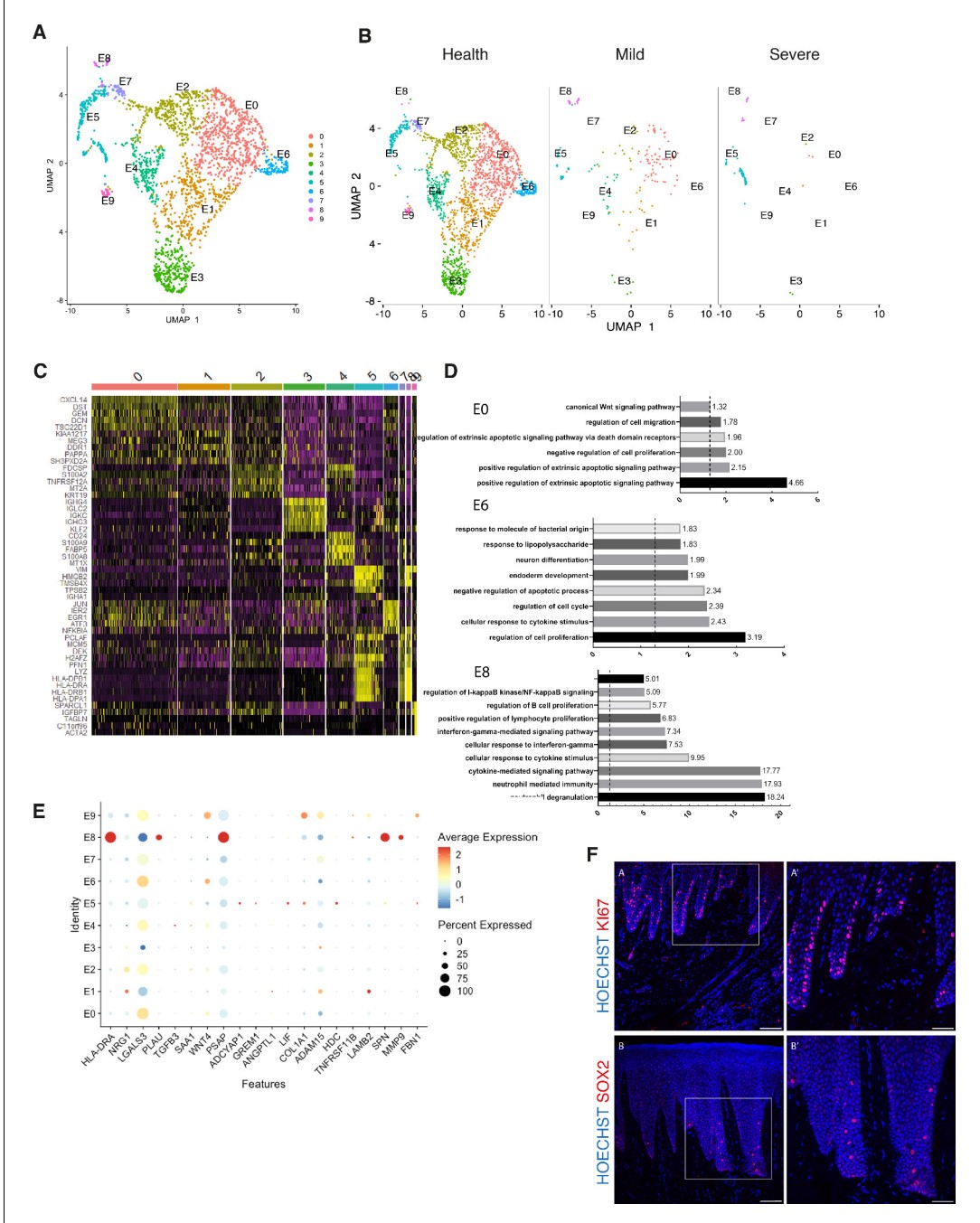

**Figure 3.** Cellular and molecular map of the epithelial gingival compartment in health and disease. (A) UMAP plot of human gingival epithelial cells. Single cells coloured by cluster annotation. (B) UMAP plot of epithelial cells during disease progression. (C) Heatmap showing subset-specific markers. (D) GO enrichment terms for E0, E6, and E8 with -log adjusted p-value shown (dotted line corresponds to FDR = 0.05). (E) Dot plot showing top predicted ligands expressed by epithelial cells that modulate the E0 (stem) compartment. (F) Expression of KI67 and SOX2 in human healthy tissue. KI67 marks proliferative cells (cluster E5), and SOX2 marks an epithelial stem cell compartment (cluster E0). Scale bars = 100 μm (A, B). Scale bars, 50 μm (A′, B′). n = 4 patient samples/condition.

The online version of this article includes the following figure supplement(s) for figure 3:

**Figure supplement 1.** Re-clustering of human epithelial gingival cells in health and disease, Related to *Figure 3*.

predict candidate stromal precursors (*Gulati et al., 2020*; *Figure 2F*). This analysis placed sub-clusters S5 and S0 as the less differentiated subpopulations, and S1 and S3 (myofibroblasts and pericytes) as fully differentiated states (*Figure 3F*). Using this pipeline, we identified genes such as *IGHBP4* and *AEBP1* in the less differentiated states. We next defined the tissue distribution of AEBP1$^+$ cells using immunohistochemistry. We observed that the majority of these cells were concentrated near the subepithelial region (*Figure 2G*).

Overall, we demonstrate that stromal remodelling in periodontitis is heterogenous with a disruption in cell populations known to be involved in tissue repair, and a higher proportion in a pro-inflammatory cell population that could prevent disease resolution. Collectively, these observations suggest that stromal cells shape a permissive inflammatory niche.

## Cellular and molecular map of the epithelial gingival compartment in health and disease

The oral epithelium is one of the fastest renewing tissues in the human body and shows a remarkable regenerative potential. Cell division in epithelial cells takes place in the basal layer which contains the stem cell compartment. After dividing, the committed cells undergo differentiation that leads to expression of structural keratins as cells move superficially (*Blanpain and Fuchs, 2009*). Recent work has started to elucidate epithelial heterogeneity in the basal layer using mouse models (*Jones et al., 2019*; *Byrd et al., 2019*). However, little is known about human gingival epithelial cell heterogeneity and its role in disease. Thus, we further explored the single-cell transcriptomes of epithelial clusters (1, 8, and 12).

By re-clustering the epithelial cells, we identified ten populations (*Figure 3A*; *Figure 3—figure supplement 1*). Two basal cell populations were identified in E0 and E1. E0 shows expression of *HOPX* which marks known stem cells in the intestinal and skin epithelia (*Takeda et al., 2013*; *Takeda et al., 2011*) and *IGFBP5* which is enriched in transit-amplifying cells (TACs) in the interfollicular epidermis (*Tumbar et al., 2004*) and recently shown through lineage-tracing to label oral epithelial stem cells in the hard palate (*Byrd et al., 2019*). E1 indicated a more mature basal cell state with expression of *DDR1* known as a cell surface receptor for fibrillar collagen, and *COL17A1*. Cycling basal cells were identified in E5 by expression of *MKI67* and *AURKB*. E2 showed enrichment for *SAA1* and *TNFRSF21* both involved in chronic inflammatory conditions. E3 showed enrichment for B-cell receptor signalling pathway, and E4 and E8 for neutrophil mediated immunity. We further identified E6 and E7 with a role in cell cycle regulation. Finally, E9 had a gene expression profile consistent with a role in ECM organisation and angiogenesis (*Figure 3A,D*; *Figure 3—figure supplement 1*).

Next, we investigated changes in epithelial cell composition and gene expression through the different disease states (*Figure 3B*). In the mild stage, we observed a depletion in E6 and E7 populations which show enrichment in genes involved in cell cycle regulation; and in E9 which is involved in ECM organisation. Cycling cells (E5) show a decrease in mild, and a subsequent increase in severe (*Figure 3B*). We detected an increase in E8 defined in GO terms by 'cytokine-mediated signalling' (*Figure 3B,D*). Next, we asked which epithelial signals are predicted to modulate the identified stem cell signature found in E0 in disease. Using NicheNet (*Browaeys et al., 2020*) we identified sub-cluster E8 as the main signalling source predicted to modulate E0 through the expression of several ligands including *MMP9*, *SPN*, and *HLA-DRA* (*Figure 3E*). While more work is necessary to understand the functional role of the E8 subpopulation, targeting this subpopulation in future immune-modulatory experiments may lead to important findings.

## Identifying ligand–receptor interactions and transcriptional regulation contributing to disease progression

Periodontitis is characterised by tissue remodelling, which depends on complex interactions between stromal, epithelial, and immune cells. However, how these cells interact to contribute to tissue homeostasis and how these interactions are dysregulated during disease remains poorly defined. To understand this cross-talk, we used NicheNet (*Browaeys et al., 2020*) to model which cellular signals induce a stromal and perivascular response in periodontitis (*Figure 4A*).

In healthy and mild stages, the cell-stromal/cell-perivascular interaction landscape was dominated by endothelial, stromal, macrophage, and epithelial originating signals (*Figure 4B*). As disease

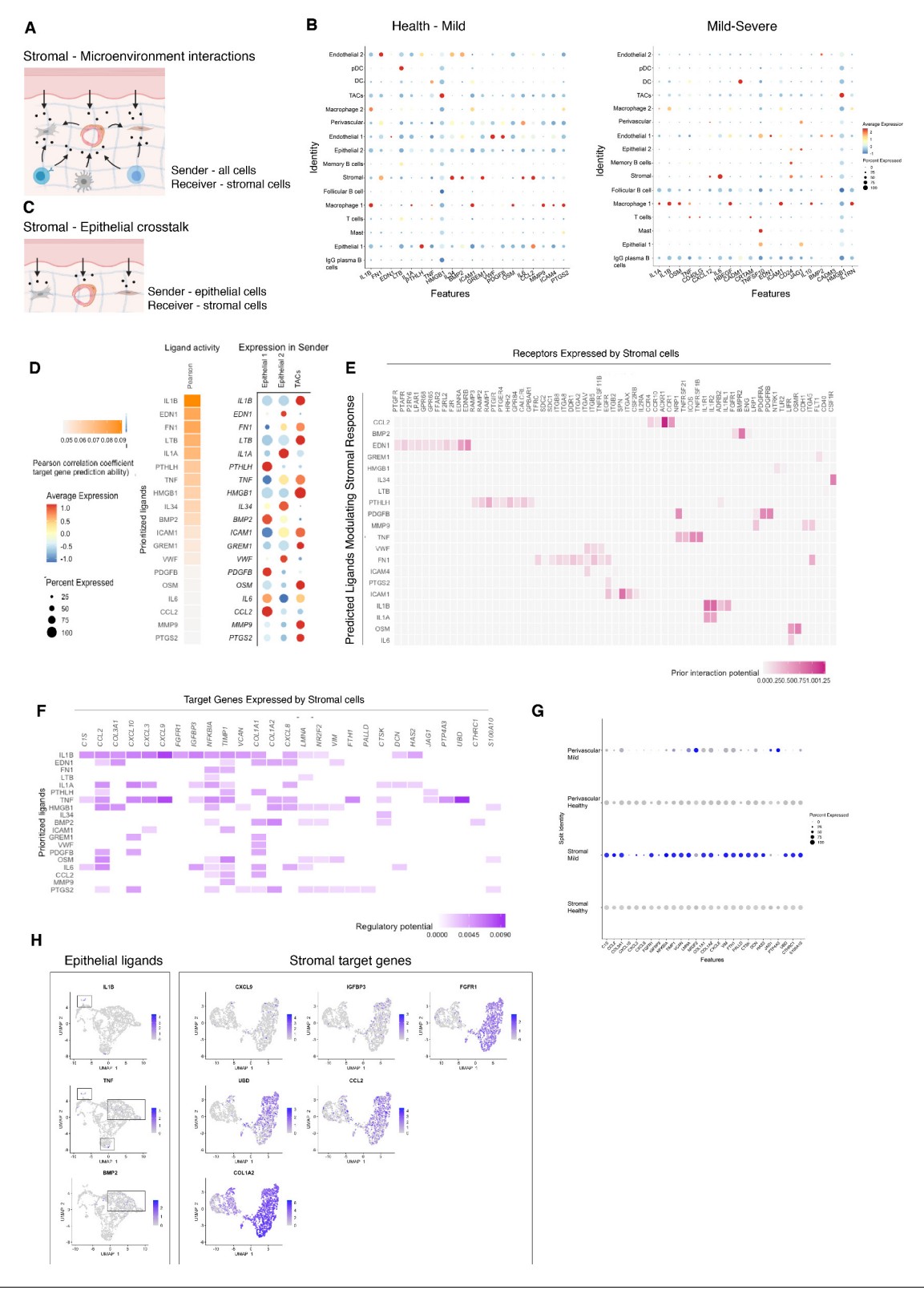

**Figure 4.** Unbiased cell–cell interaction analysis and its effect in the stromal microenvironment. (**A**) Schematic representation of the NicheNet analysis of upstream ligand–receptor pairs and stromal target genes inducing DE genes in periodontitis. Created with BioRender.com. (**B**) Dot plots depicting which gingival cell populations express top-ranked ligands contributing to the transcriptional response observed from health to mild disease and from mild to severe in the stromal compartment. (**C**) Schematic representation of the NicheNet analysis of epithelial-mesenchymal crosstalk in mild disease.
*Figure 4 continued on next page*

*Figure 4 continued*

Created with BioRender.com. (D) Top predicted epithelial ligands driving the stromal inflammatory response and dot plot showing which epithelial subpopulation express these ligands. (E) Ligand-receptor heatmap of potential receptors expressed by stromal cells associated with each epithelial ligand. (F) Ligand-target heatmap of stromal and perivascular target genes of the identified epithelial ligands. (G) Dot plot confirming upregulation of the identified stromal target genes in disease. (H) UMAPs feature plots mapping the identified epithelial ligands and target genes to the respective target genes expressed by stromal cells.

progresses, in mild and severe stages, we observed a clear loss in endothelial and stromal originating signals, and an increase in macrophage, mast, T, and B-cell signalling (*Figure 4B*). Analysis of these cell–cell interactions revealed several signalling pathways including tumour necrosis factor (TNF) and bone morphogenetic protein (BMP) signalling (*Figure 4B*). Overall, the number of predicted interactions in severe disease was strongly reduced.

We next focused on epithelial–mesenchymal interactions in the mild stage by investigating which signalling interactions could potentially induce an inflammatory signature in the mesenchymal compartment (*Figure 4C*). Analysis of epithelial ligands predicted to cause an inflammatory response revealed *IL1*, *EDN1*, *TNF*, *LTB*, and *BMP2* as the main contributors to the mild inflammatory stage (*Figure 4D*). Proliferative cells (TACs) are suggested to be the main source of these ligands with the exception of *BMP2* (*Figure 4D*). We next analysed which stromal and perivascular receptors can potentially bind to these identified epithelial ligands (*Figure 4E*) and the target genes of these ligand–receptor interactions (*Figure 4F*). We estimated prominent *IL1B-CXCL9*, *TNF-CXCL9*, *TNF-UBD*, *BMP2-COL1A2* interactions, suggesting that these molecular interactions may be crucial in sustaining a proinflammatory microenvironment. Target genes were confirmed to be differentially expressed with disease (*Figure 4G*). *IL1B* and *TNF* epithelial ligands specifically targeted S0 and S5 stromal subpopulations, and *BMP2* all fibroblast-like subpopulations and pericytes (*Figure 4H*).

Together, these results identify *IL1B*, *EDN1*, *TNF*, and *BMP2* as the main epithelial modulators driving an inflammatory response in stromal and perivascular cells. Based on their expression, we identified novel epithelial-mesenchymal interactions in periodontitis: the interactions between epithelial *IL1B* and *TNF* and stromal target genes. It is unclear if the interactions described above are causative or consequential of disease progression; however, this analysis provides a wealth of novel targets that can be pharmacologically targeted in studies aiming to ameliorate the disease phenotype.

## Single-cell transcriptomics of human B cells reveals activation signature in periodontitis

B cells are essential in the generation of protective immunity. However, tissue-based B-cell subsets are not well characterised in human oral tissues. Following our observations that there is a consistent increase of B cells in line with disease severity, and their established role in disease immunopathogenesis, we next focused on the humoral response by performing a more in-depth transcriptomic analysis. Previous studies have established that B cells constitute the majority of cells in periodontitis lesions (*Thorbert-Mros et al., 2015*), and it has been suggested a dual protective and detrimental role (*Oliver-Bell et al., 2015*; *Abe et al., 2015*).

We compared their transcriptional profiles across disease states (*Figure 5A*). We found a profound prevalence of IgG plasma B cells in disease which is supported by another study (*Kinane et al., 1999*) in periodontitis patients. Similarly, it has been reported an increase in local IgG within the gastrointestinal tract during intestinal inflammation (*Castro-Dopico et al., 2019*). Here, we found IgG plasma cells almost absent in health and distinctively expanded with disease progression (*Figure 5A*). Upregulation of an *IGH* signature has been previously linked to disease severity and renders activation of the mononuclear phagocyte response in the intestinal mucosa (*Castro-Dopico et al., 2019*). In humans, mucosal IgG responses are pro-inflammatory when they involve complement activation. This cluster showed enrichment of genes involved in the complement system such as CFB and C2 (*Figure 5D*). This system plays a critical role in signalling B-cell activation (*Carroll and Isenman, 2012*; *Chen et al., 2020*), and previous research has established a role in

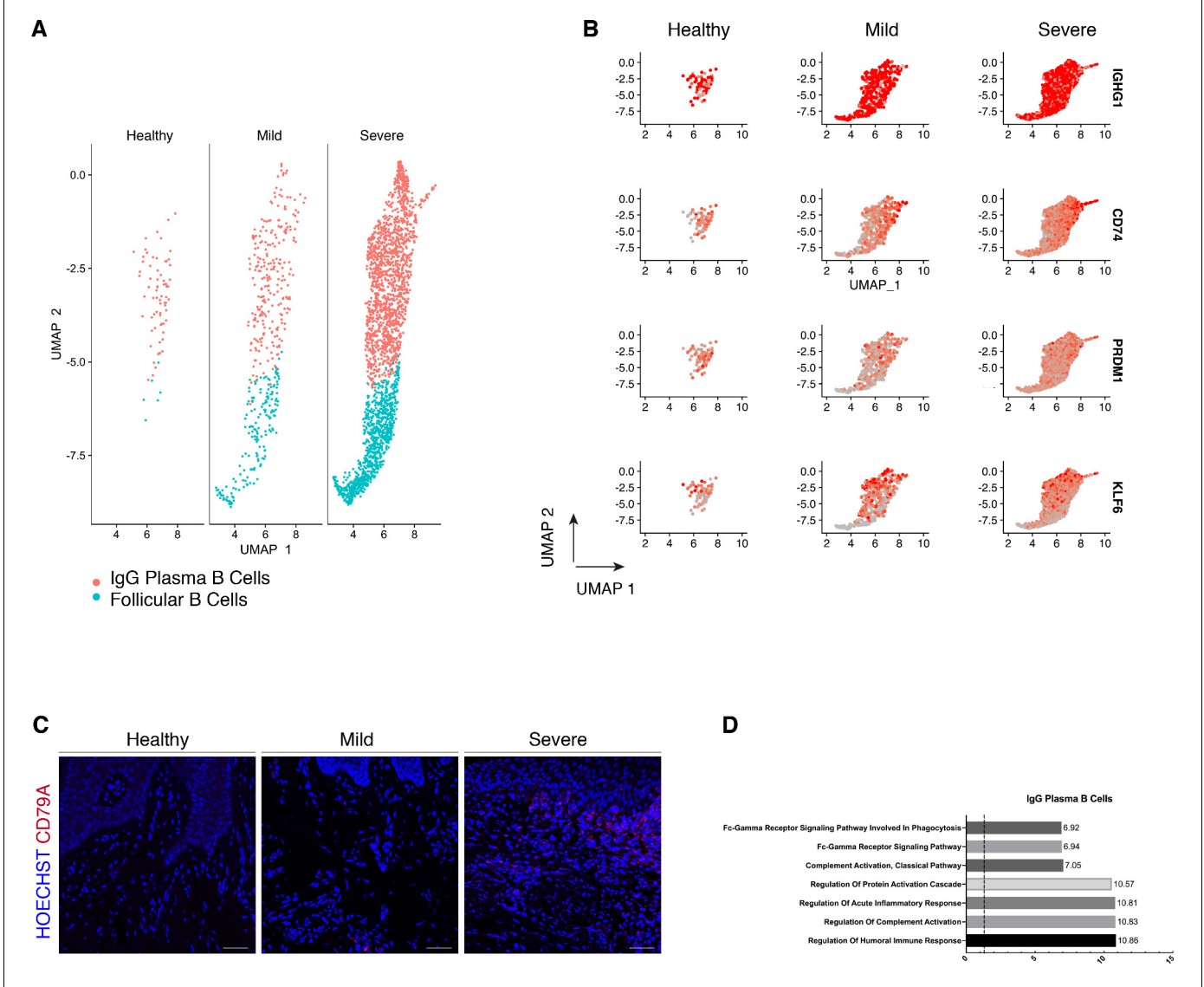

**Figure 5.** Periodontitis induces an IgG plasma B-cell signature in human gingiva. (**A**) UMAP analysis of human B cells identifying follicular and IgG plasma B cells split by condition. (**B**) UMAP expression plots of human B-cell subset markers. Cells coloured by normalised expression of indicated genes. (**C**) CD79A in human gingival tissue across health and disease. Scale bars, 100 μm. n = 3 patient samples/condition. (**D**) Gene enrichment analysis of IgG plasma B cells. -log adjusted p-value shown (dotted line corresponds to FDR = 0.05).

The online version of this article includes the following figure supplement(s) for figure 5:

**Figure supplement 1.** Re-clustering of human B and T cells in health and disease.

periodontitis. There was no inherent transcriptional heterogeneity to allow further re-clustering of these B-cell into more bioinformatically refined subtypes (*Figure 5—figure supplement 1*).

## Discussion

The human gingiva is a unique barrier site since failure to appropriately control immune responses leads to periodontitis. However, the molecular mechanisms of homeostasis and how they are disrupted in disease are poorly understood. Previous studies have reported on gene expression in

gingival tissue from patients with periodontitis, however these studies have used conventional bulk RNA sequencing on whole-biopsies which average gene expression changes across the whole tissue, and therefore lose all information of discreet cellular subpopulations (*Becker et al., 2014*; *Davanian et al., 2012*; *Demmer et al., 2008*; *Kim et al., 2016*; *Lundmark et al., 2015*). In this work, we provided the first comprehensive cellular landscape of in vivo human gingiva, charting dynamic cellular composition differences at single-cell level across disease states. Our atlas comprises all the main gingival cell types defined by the expression of canonical and novel gene markers, with highly consistent results across all samples tested. Next, we analysed the potential molecular signals driving the inflammatory response in the stromal niche.

We identified a striking difference in mesenchymal and epithelial cells during disease progression. In the mesenchymal lineage, we identified populations of established cells, such as myofibroblasts and pericytes, and five additional distinct populations of fibroblast-like cells. Recent studies have started to elucidate the role of stromal cell populations in tissue homeostasis (*Shoshkes-Carmel et al., 2018*; *Bahar Halpern et al., 2020*; *Greicius et al., 2018*), and consistent with previous studies we identified two populations expressing Wnts and Wnt inhibitors suggesting the presence of mesenchymal niche regulating populations (*Kinchen et al., 2018*; *Kim et al., 2020*) that may be required for oral mucosa maintenance. We also identified a population defined by AEBP1 expression concentrated in the subepithelial region and associated with a less differentiated stromal state. This protein has been reported to be a DNA-binding transcriptional repressor with role in smooth muscle differentiation and wound healing (*Layne et al., 2001*). In periodontitis, we observed a selective loss of stromal populations; fibroblast-like cells were preserved in the mild stage, whereas the myofibroblast and pericyte populations were strikingly reduced. Myofibroblasts are known to be responsible for excessive synthesis, deposition, and remodelling or extracellular matrix proteins (*Tomasek et al., 2002*), however less is known about the mechanisms that promote their survival and persistence in inflammatory conditions. Multiple single-cell analysis have revealed that myofibroblast populations are heterogenous and undergo dynamic changes during tissue repair in various organs (*Farbehi et al., 2019*; *Guerrero-Juarez et al., 2019*; *Xie et al., 2018*; *Tabib et al., 2018*; *Peyser et al., 2019*; *Lambrechts et al., 2018*). Our observation that myofibroblasts are reduced in the transition from health to mild disease, might suggest a contribution to ECM degradation and to the persistence of a chronic inflammatory state characteristic of periodontitis. Previous research has suggested two mechanisms that limit myofibroblast survival; either a dependence on growth factor receptor-mediated pathways required for their survival (*Boström et al., 1996*), or pro-apoptotic cytokines might selectively induce apoptosis by directly activating cell death signalling pathways or by inhibiting pro-survival pathways. One example is IL-1B which induces caspase-dependent apoptosis in mouse lung myofibroblasts by inhibiting FAK (*Zhang and Phan, 1999*). We also detected a decrease in the pericyte population from health to mild disease. Pericytes are present in all vascularised tissues, and provide structural support to the vasculature with proven roles in angiogenesis (*Lindblom et al., 2003*), wound healing (*Kramann et al., 2015*), progenitor cell functions (*Crisan et al., 2008*) and immunomodulation (*Meyers et al., 2018*; *Yianni and Sharpe, 2020*). It has been demonstrated that there is an expansion and dilation of the vasculature in periodontitis (*Zoellner et al., 2002*), contributing to increased leukocyte recruitment into the tissue. The loss or detachment of pericytes has been implicated in disease (*Armulik et al., 2011*), and has been related to infiltration of inflammatory cells (*Ogura et al., 2017*). Interestingly, *Pdgfb* or *Pdgfr* loss-of-function embryos show vascular hyperplasia and microvessel dilation (*Hellström et al., 2001*). We hypothesise that the observed pericyte decrease might impair the stromal compartments ability to regenerate as these are mesenchymal stem cell precursors in vivo (*Sacchetti et al., 2016*; *Yianni and Sharpe, 2018*).

In periodontitis, we observed the emergence of one fibroblast-like population highly enriched in pro-inflammatory genes such as AREG. Overall, we observed stromal remodelling in a subpopulation specific way and in accordance with previous reports (*Kinchen et al., 2018*). Normal repair and regeneration responses are compromised, while continuous production of pro-inflammatory factors prevent inflammatory resolution.

This work also provides the first comprehensive analysis of the human gingival epithelium. Understanding the molecular mechanisms underlying this mucosal barrier can help shape immunoregulatory responses in the context of homeostasis and disease. Our data identified a basal progenitor cell population expressing *HOPX* and *IGFBP5*. Although, recent studies have started to elucidate oral

progenitor cells' heterogeneity, this is the first human detailed characterisation that will allow the development of future validation models. We identified one epithelial subpopulation (E8) expanded in disease, and intercellular communication analysis suggested that this population is the main signalling centre driving the epithelial inflammatory response. More work is needed to address this finding and the immunoregulation of this population.

We provided an immune repertoire profiling and described in detail the expansion of B-cell subtypes. These results are consistent with data obtained in a previous study despite the difference in tissue collection. Our samples were obtained from sites which had received non-surgical treatment but still had residual disease and the Dutzan study collected from a cohort that had never been treated for disease (*Dutzan et al., 2016*). We also observed a T-cell-rich inflammatory infiltrate with minimal B cells present in health. This rich and diverse immune network present in health explains the immunosurveillance required to control the constant bacterial exposure. While Dutzan et al, identified neutrophils as the most notable cellular difference in periodontitis we were unable to capture these cells due to the digestion protocol used to dissociate these biopsies. single-cell analysis of neutrophils has known technical challenges due to their low RNA content, high level of RNases and high susceptibility to degradation during tissue digestion. Hence, we acknowledge this as an important study limitation of the immune compartment. In addition, we seem to be capturing a reduced number of T-cells which we were able to only re-cluster into CD4 and CD8 T-cell subtypes (*Figure 5—figure supplement 1*). This sequencing limitation specific to T-cells has been acknowledged by independent studies showing that T-cells are difficult to bioinformatically segregate and re-cluster unless sequenced to high numbers and depth (*Ding et al., 2020*).

We provided a detailed molecular description of B-cell subsets as it was the major cellular shift detected in the immune cell network. This is consistent with previous observations showing that the most upregulated genes in periodontitis are involved in B-cell development (*Lundmark et al., 2018*). Despite the knowledge that atypical activation of B cells contribute to disease progression by their antigen-presentation, cytokine production, and expression and secretion of receptor activator of nuclear factor-kB ligand (RANKL), contributing to osteoclastogenesis (*Thorbert-Mros et al., 2015*), little is known about the molecular mechanisms driving these processes. We identified a specific IgG plasma cell response. Recently, a IgG contribution has been specifically linked with driving chronic inflammatory responses (*Castro-Dopico et al., 2019*). In that study, patient samples with higher levels of IgG had the highest disease severity scores and correlated with neutrophil infiltration and IL-1B expression. In our study, this response was associated with complement activation. Previously, complement split products were found absent or present at low concentrations in healthy individuals, but abundant in periodontitis (*Damgaard et al., 2015*; *Hajishengallis et al., 2017*). Continuous complement activation promotes survival of local pathogens in a nutritionally favourable inflammatory environment that promotes dysbiosis and disease development (*Hajishengallis et al., 2017*; *Hajishengallis et al., 2011*; *Maekawa et al., 2014*). While we could not bioinformatically resolve multiple B-cell subtypes, our findings have therapeutic implications by identifying IgG signalling as a potential therapeutic target in periodontitis.

Finally, we aimed to identify the signals driving the inflammatory response in the stromal compartment. Previous studies have reported IL1B and TNF as key regulators in the periodontitis pathogenesis (*Yucel-Lindberg and Båge, 2013*), therefore it was not surprising to find these molecules highly represented in our cell interaction analysis (*Figure 4*). We described newly identified molecular mechanisms involved in the regulation of these cytokines by predicting new receptor interactions and previously unidentified target genes. These findings bring new perspectives on periodontitis molecular mechanisms governing tissue loss and future experiments will be important to test these predictions.

In summary, we have established the first human gingiva cell atlas, revealing heterogeneity within major gingiva cell populations and providing with a roadmap for further functional insights into the immune and structural populations present in the gingiva. It also provides new biological insights into the immunopathogenesis of periodontitis. These data offer enormous potential for medicine, drug discovery and diagnostics through a more detailed understanding of cell types, basic biological processes and disease states.

# Materials and methods

## Key resources table

| Reagent type (species) or resource | Designation | Source or reference | Identifiers | Additional information |
|---|---|---|---|---|
| Biological sample (Human) | Human gingival biopsies | Periodontology department, King's College London | | |
| Antibody | anti-COLVI (Rabbit monoclonal) | Abcam | Cat #ab182744, RRID:AB_2847919 | IHC (1:500) |
| Antibody | anti-ACTA2 (Mouse monoclonal) | Abcam | Cat #ab7817, RRID:AB_262054 | IHC (1:200) |
| antibody | anti-MCAM (Rabbit monoclonal) | Abcam | Cat #ab75769, RRID:AB_2143375 | IHC (1:100) |
| Antibody | anti-KI67 (Rabbit polyclonal) | Abcam | Cat #ab15580, RRID:AB_443209 | IHC (1:100) |
| Antibody | anti-SOX2 (Rabbit monoclonal) | Abcam | Cat# ab92494, RRID:AB_10585428 | IHC (1:100) |
| Antibody | anti-CD79A (Rabbit monoclonal) | Abcam | Cat# ab79414, RRID:AB_2260147 | IHC (1:100) |
| Antibody | anti-AEBP1 (Rabbit polyclonal) | Atlas Antibodies | Cat# HPA064970, RRID:AB_2685394 | IHC (1:200) |
| Commercial assay or kit | ImmPRESS Excel Staining Kit, Anti-Rabbit Ig | Vector Laboratories | Cat# MP-7601, RRID:AB_2336533 | |
| Chemical compound, drug | UltraPure BSA (50 mg/mL) | ThermoFisher Scientific | Cat# AM2618 | 0.04% |
| Commercial assay or kit | Chromium Single Cell 3' Library and Gel Bead Kit v3 | 10X Genomics | Cat# PN-1000092 | |
| Commercial assay or kit | Chromium Single Cell B Chip Kit | 10X Genomics | Cat# PN-1000074 | |
| Commercial assay or kit | Whole Skin Dissociation Kit, human | Miltenyi Biotec | Cat# 130-101-540 | |
| Software, algorithm | CellRanger Version 4 | 10X Genomics | RRID:SCR_017344 | |
| Software, algorithm | Seurat Version 3.0 | R Bioconductor | RRID:SCR_007322 https://satijalab.org/seurat/ | |
| Software, algorithm | Enrichr | *Chen et al., 2013* | RRID:SCR_001575 | |
| Software, algorithm | CytoTRACE | R Bioconductor | https://cytotrace.stanford.edu | |
| Software, algorithm | NicheNet | GitHub | https://github.com/saeyslab/nichenetr | |
| Other | GRCh38 | CellRanger, 10X Genomics | https://support.10xgenomics.com/single-cell-gene-expression/software/downloads/latest | |
| Other | DAPI stain | Invitrogen | D1306, RRID:AB_2629482 | (1 ug/mL) |

## Patient recruitment and ethical approval

Human gingival samples were obtained from consenting patients undergoing routine periodontal surgical procedures (Department of Periodontology, Guy's Hospital, King's College London). All samples were collected and processed in compliance with the UK Human Tissue Act (Human Tissue Authority #203019), ethically approved by the UK National Research Ethics Service (Research Ethics Committee 17/LO/1188). Written informed consent was received from participants prior to inclusion in the study. Cohort inclusion criteria for all subjects were: absent history of relevant medical conditions, no use of medication, no use of nicotine or nicotine-replacement medications, no pregnancy, and breast feeding.

Healthy controls included crown lengthening procedures, and periodontitis patients, pocket reduction surgeries. Patients with periodontitis had tooth sites with probing depth $\geq 6$ mm, and bleeding on probing. Patients used as controls showed no signs of periodontal disease, with no gingival/periodontal inflammation, a probing depth $\leq 3$ mm, and no bleeding on probing.

Patient 33. Gender: male. Age band: 41–65. No history of periodontal disease. Site: buccal gingival margin.

Patient 35. Gender: female. Age band: 41–65. Chronic periodontitis with previous history of nonsurgical treatment (mild). Site: buccal gingival margin.

Patient 37. Gender: male. Age band: 41–65. Chronic periodontitis with previous history of nonsurgical treatment (severe). Site: buccal gingival margin.

Patient 38. Gender: male. Age band: 41–65. No history of periodontal disease. Site: buccal gingival margin.

## Histology and microscopy

Human gingival tissue was freshly collected and fixed overnight in 4% neutral buffered formalin. Then, tissue underwent three 5 min washes in PBS at room temperature followed by dehydration washes in increasing ethanol concentrations. After dehydration, tissue was processed using a Leica ASP300 Tissue Processing for one hour. Tissues were then embedded in paraffin. Serial sections (12 µm thick) were cut for haematoxylin and eosin (H and E) and immunohistochemistry (IHC) staining.

H and E was carried out for each patient sample using an Automated Slide Stainer. Slides were dewaxed by immersion in Neo-Clear (Merck Millipore), twice for 10 min. Tissue was then rehydrated by decreasing volumes of ethanol in deionised $H_2O$ (100, 90, 70, 50%) for two minutes in each step and rinsed in deionised $H_2O$ for 2 min. Samples were then stained in Ehrlich's Haematoxylin (Solmedia) for 10 min followed by a 10 min rinse under running water and then a two-minute rinse in deionised $H_2O$. Tissue was then stained in 0.5% Eosin Y (Sigma-Aldrich) for 5 min and washed twice in deionised $H_2O$. Samples were dehydrated in increasing IMS in deionised $H_2O$ concentration steps (70, 90, 100, 100%) for two minutes each. Slides were immersed in Neo-Clear three times for 5 min and then mounted using Neo-mount mounting medium (Merck Millipore), coverslipped and left to dry overnight in at 42℃.

## Immunohistochemical staining

Immunofluorescence staining was performed on 12 µm sections as described above. In short, slides were dewaxed in Neo-Clear twice for 10 min and rehydrated in a series of decreasing ethanol volumes as described above. Heat induced epitope retrieval was performed with sodium citrate buffer (pH 6) in a Decloaking chamber NXGEN (Menarini Diagnostics) for 3 min at 110℃. Slides were cooled to room temperature before blocking for 1 hr at room temperature in Blocking Buffer (0.2% BSA, 0.15% glycine, 0.1% TritonX in PBS) with 10% goat or donkey serum depending on the secondary antibody used. Primary antibodies were diluted in blocking buffer with 1% of the respective blocking buffer and incubated overnight at 4℃. The following day, slides were washed three times in PBST and incubated with the respective secondary antibodies diluted 1:500 in 1% blocking buffer for one hour at room temperature. Slides were mounted with Citifluor AF1 mountant media (Citifluor Ltd., AF1-100) and cover slipped for microscopy. Slides were put to dry in a dry chamber that omitted all light, and kept at 4℃. For AEBP1 staining, slides were incubated following primary antibody step with ImmPRESS Excel Staining Kit Anti-Rabbit IgG (Peroxidase) Polymer Detection Kit (Vector Laboratories, Peterborough, U.K.) for 30 min at room temperature. Peroxidase activity was visualised using ImmPACT DAB Peroxidase (HRP) Substrate (Vector Laboratories). Finally, sections were

counter-stained with Mayer's hematoxylin, dehydrated, and mounted. Primary antibody was excluded from negative controls.

The following antibodies were used: COLVI raised in rabbit (ab182744, 1:500, Alexa Fluor-488 secondary), ACTA2 raised in mouse (ab7817, 1:200, Alexa Fluor-633), MCAM raised in rabbit (ab75769, 1:100, Alexa Fluor-594), KI-67 raised in rabbit (ab15580, 1:100, Alexa Fluor-594), SOX2 raised in rabbit (ab92494, 1:100, biotinylated secondary), CD79A raised in rabbit (ab79414, 1:100, Alexa Fluor −488 secondary), AEBP1 raised in rabbit (HPA064970, 1:200).

## Imaging

For bright field images, stained slides were scanned with Nanozoomer-XR Digital slide scanner (Hamamatsu) and images processed using Nanozoomer Digital Pathology View. Fluorescent staining was imaged with a TCS SP5 confocal microscope (Leica Microsystems) and Leica Application Suite Advanced Fluorescence (LAS-AF) software. Images were collected and labelled using Adobe Photoshop 21.1.2 software and processed using Fiji (*Schindelin et al., 2012*).

## Tissue processing for single-cell isolation

Fresh tissues were processed immediately after clinical surgery using the same protocol. Tissue was transferred to a sterile petri dish and cut into <1 mm³ segments before being transferred to a 15 mL conical tube. Tissue was digested for 30 min at 37°C with intermittent shaking using an enzymes dissociation kit (Miltenyi, Bergisch-Gladbach, Germany). The resulting cell suspension was filtered through a 70 μm cell strainer to ensure a single-cell preparation and cells collected by centrifugation (1,200 rpm for 5 min at 4°C). Cells were resuspended in 0.04% non-acetylated BSA (UltraPure BSA, ThermoFisher Scientific) and stained with 1.5 μg DAPI (D1306, Invitrogen) used as dead cell exclusion marker. Samples were analysed on BD FACS Aria III fusion machine. Cells were gated based on size using standard SSC-A and FSC-A parameters so that debris is excluded. Doublets were excluded using SSC-A and SSC-W parameters. Live cells were selected as cells identified to be dimly fluorescing in DAPI and were then sorted into chilled FACS tubes prefilled with 0.04% 300 μL BSA. Single-cell suspensions were captured using the 10X Genomics Chromium Single Cell 3' Solution (v3) according to the manufacturers protocol. Cells were resuspended separately in PBS with 0.04% BSA at a density of 50–100 cells per μL. While neutrophils were not specifically targeted for exclusion, due to their small size they heavily co-localised with cellular debris in FACS plots, and therefore not included. Isolation of neutrophils for single-cell RNA sequencing on Chromium platforms has been thoroughly documented and reported (*Smillie et al., 2019*).

## Chromium 10x genomics library and sequencing

Single-cell suspensions were manually counted using a haemocytometer and concentration adjusted to a minimum of 300 cells μL$^{-1}$. Cells were loaded according to standard protocol of the Chromium single-cell 3' kit to capture around 5,000 cells per chip position. Briefly, a single-cell suspension in PBS 0.04% BSA was mixed with RT-PCR master mix and loaded together with Single Cell 3' Gel Beads and Partitioning Oil into a Single Cell 3' Chip (10x Genomics) according to the manufacturer's instructions. RNA transcripts from single cells were uniquely barcoded and reverse transcribed. Samples were run on individual lanes of the Illumina HiSeq 2500.

## Computational analysis of sc-RNAseq datasets

The cell ranger pipeline was used for processing of the single-cell RNAseq data files prior to analysis according to the instructions provided by 10x Genomics. Briefly, base call files obtained from each of the HiSeq2500 flow cells used were demultiplexed by calling the 'cellranger mkfastq'. Resulting FASTQ files were aligned to the human reference genome GRCh38/hg19 and subsequently filtered and had barcodes and unique molecular identifiers counted and count files generated for each sample. These were used for subsequent processing and data analysis using R.

## Integrative analysis

Integrated analysis was performed according to the authors of the Seurat package (*Butler et al., 2018*; *Stuart et al., 2019*). Briefly, count files for each condition were read into RStudio and cells corresponding to each condition were labelled accordingly as 'Healthy', 'Mild' and 'Severe'. Only

cells found to be expressing more than 500 transcripts were considered as to limit contamination from dead or dying cells. Each dataset was normalised for sequencing depth by calling the 'NormalizeData' function and the 2000 most variable features of each dataset were detected using the 'vst' method by calling the 'FindVariableFeatures' function. Subsequently the 'FindIntegrationAnchors' function was called to identify anchors across the datasets and the 'IntegrateData' function to integrate them so an integrated analysis could be run on all cells simultaneously. The data was then scaled to account for sequencing depth using 'ScaleData' and PCA components were used for an initial clustering of the cells (using 'RunPCA'). 20 dimensions were used to capture the majority of the variability across the datasets. 'FindNeighbors' was then used, utilising the above dimensionality parameters to construct a K-nearest neighbour graph based on Euclidian distances in PCA space. The clusters are then refined by applying a Louvain algorithm that optimises the modularity of the dataset and groups the cells together based on global and local characteristics. This is done by calling the 'FindClusters' function. We then run non-linear dimensionality reduction using UMAP to be able to visualise and explore the datasets. The same principle components were used as above. The Stromal, Epithelial, and B-cell clusters were then extracted using the 'Subset' function.

## Stromal cell re-clustering analysis

Stromal clusters were identified as being 'collagen producing'. These two clusters were reanalysed separately from the integrated dataset. Stromal cells were filtered to only utilise live cells using percentage of mitochondrial gene expression as an exclusion metric (<15%). Datasets were then re-normalised by calling the 'NormalizeData' function to account for the reduction in cell numbers subsequent to subsetting the data. According to the author instructions, the top 2000 most variable features across the dataset were then identified using the 'FindVariableFeatures'. These variable features were subsequently used to inform clustering by passing them into the 'RunPCA' command. Using 'Elbowplot' we identified that the first eight principle components should be used for downstream clustering when invoking the 'FindNeighbors' and 'RunUMAP', as detailed above.

## Epithelial cell re-clustering

Epithelial cells were identified from the epithelial clusters in the integrated UMAP and re-clustered as explained above with some minor exceptions. Epithelial cells were isolated as being the clusters 1, 8, 12. The first five principal components were used as these were identified as being significant by the 'Elbowplot' function.

## Gene Ontology (GO) analysis

Gene ontology (GO) analysis was performed using Enrichr (*Chen et al., 2013*) on the top 200 differentially expressed genes (adjusted p-value < 0.05 by Wilcoxon Rank Sum test). GO terms shown are enriched at FDR < 0.05.

## CytoTRACE

An expression matrix consisting of only the specified sub-set of cellular populations was used as a starting point. CytoTRACE analysis was performed according to the developer's instructions (*Gulati et al., 2020*). The resulting embeddings were then projected onto the UMAP projections.

## NicheNet analysis

This analysis predicts which ligands produced by a sender cell regulate the expression of receptors/ target genes in another (receiver) cell. We followed the open-source R implementation available at GitHub (https://github.com/saeyslab/nichenetr). For differential expression we used FindMarkers function in Seurat to generate average logFC values per cell type. For *Figure 3E*, we assigned all epithelial populations as 'sender cells' and E0 as 'receiver' to derive a set of predicted epithelial ligands modulating the mild response seen in this specific subpopulation. For *Figure 4B*, we assigned all cell types as 'sender cells' and the stromal populations as 'receiver' and extracted all cell type signatures by taking the 100 differentially expressed genes isolated in health/mild and in mild/severe.

For *Figure 4D–G*, we defined all epithelial populations as 'sender' and all stromal as 'receiver' in health vs mild responses.

## Study approval

Informed consent in writing before their participation in this study was obtained from each subject in compliance with the UK Human Tissue Act (Human Tissue Authority #203019), and ethically approved by the UK National Research Ethics Service (Research Ethics Committee 17/LO/1188).

## Acknowledgements

We thank all the patients who contributed to this study, the support of our Periodontology MClinDent students, GSTT nursing staff and Dr Pegah Heidarzadeh Pasha at Guy's Hospital. We thank all CCRB laboratory technicians for their support, especially Dr Alasdair Edgar for tissue processing and H and E staining, and Dr Susmitha Rao for tissue culture. We acknowledge support of the BRC Flow Cytometry and BRC Genomics cores at Guy's Hospital for their services. We thank Dr Ranjit Bhogal, Dr Fei Ling Lim, Miss Alison Russell and Dr Jenny Pople for their support at Unilever. The research described was supported by the BBSRC Industrial CASE Studentship (Grant Ref: BB/P504506/1) and National Institute for Health Research's Biomedical Research Centre based at Guy's and St Thomas' NHS Foundation Trust and King's College London. The views expressed are those of the authors and not necessarily those of the NHS, the National Institute for Health Research, or the Department of Health. This work was funded by Unilever in the form of a research grant awarded to PTS.

The authors state no conflict of interest. However, for the record, ED'A is an employee of Unilever Plc.

## Additional information

### Competing interests

Eleanor M D'Agostino: an employee of Unilever Plc. Paul Sharpe: This work was funded by Unilever in the form of a research grant awarded to P.T.S. The other authors declare that no competing interests exist.

### Funding

| Funder | Grant reference number | Author |
| --- | --- | --- |
| Biotechnology and Biological Sciences Research Council | BB/P504506/1 | Paul Sharpe |
| NIHR | | Paul Sharpe |
| Guy's and St Thomas' NHS Foundation Trust | | Paul Sharpe |

The funders had no role in study design, data collection and interpretation, or the decision to submit the work for publication.

### Author contributions

Ana J Caetano, Data curation, Formal analysis, Validation, Investigation, Methodology, Writing - original draft, Writing - review and editing; Val Yianni, Data curation, Software, Formal analysis, Writing - original draft, Writing - review and editing; Ana Volponi, Resources, Validation, Writing - review and editing; Veronica Booth, Resources; Eleanor M D'Agostino, Conceptualization, Supervision, Project administration, Writing - review and editing; Paul Sharpe, Conceptualization, Supervision, Funding acquisition, Writing - original draft, Project administration, Writing - review and editing

### Author ORCIDs

Ana J Caetano (iD) https://orcid.org/0000-0003-4588-3241
Val Yianni (iD) https://orcid.org/0000-0001-9857-7577
Paul Sharpe (iD) https://orcid.org/0000-0003-2116-9561

## Ethics

Human subjects: Informed consent in writing before their participation in this study was obtained from each subject in compliance with the UK Human Tissue Act (Human Tissue Authority #203019), and ethically approved by the UK National Research Ethics Service (Research Ethics Committee 17/LO/1188).

## Decision letter and Author response

Decision letter https://doi.org/10.7554/eLife.62810.sa1
Author response https://doi.org/10.7554/eLife.62810.sa2

# Additional files

## Supplementary files

- Source data 1. Cluster markers (all, stromal, epithelial).
- Source data 2. Gene Set Enrichment Analyses (Stromal).
- Source data 3. Gene Set Enrichment Analyses (Epithelial).
- Transparent reporting form

## Data availability

Raw sequencing data obtained from patients used in this study is deposited under GSE152042.

The following dataset was generated:

| Author(s) | Year | Dataset title | Dataset URL | Database and Identifier |
|---|---|---|---|---|
| Caetano A, Yianni V, Volponi A, Booth V, D'Agostino EM, Sharpe P | 2020 | Defining human mesenchymal and epithelial heterogeneity in response to oral inflammatory disease | http://www.ncbi.nlm.nih.gov/geo/query/acc.cgi?acc=GSE152042 | NCBI Gene Expression Omnibus, GSE152042 |

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
