## [Decision Letter]

**Acceptance summary:**

In this study, the authors investigated the cellular heterogeneity of human oral mucosa in health and disease using single cell transcriptome profiling, and identified potential ligand-receptor interaction between different cell types that drive the inflammation of periodontitis. Overall, the concept of the study is of great interest and human data from healthy and diseased tissue may provide valuable information to the field. The study has been improved with some in vivo validation and clarification.

**Decision letter after peer review:**

Thank you for submitting your article "Defining human mesenchymal and epithelial heterogeneity in response to oral inflammatory disease" for consideration by *eLife*. Your article has been reviewed by two peer reviewers, and the evaluation has been overseen by a Reviewing Editor and Kathryn Cheah as the Senior Editor. The reviewers have opted to remain anonymous.

The reviewers have discussed the reviews with one another and the Reviewing Editor has drafted this decision to help you prepare a revised submission.

Summary:

This manuscript investigated the cellular heterogeneity of human oral mucosa in health and disease using single cell transcriptome profiling, and identified potential ligand-receptor interaction between different cell types that drive the inflammation of periodontitis. Overall, the concept of the study is of great interest and human data from healthy and diseased tissue may provide valuable information to the field.

Essential revisions:

1) For the integration analysis, the authors should state how many cells were analyzed from each healthy and periodontitis patient.

2) Using computational analysis, the authors identified a subpopulation of stromal cells expressing *IGHBP4* and *AEBP1* that are in a less differentiated state. It would be helpful to analyze some human gingival tissue and see where these AEBP1+ cells are located.

3) In the integration analysis, it seems that epithelium is first affected relative to the stromal population in periodontitis patients. Even in the mild cases, the epithelial cell number is dramatically reduced. However, histology is not echoing this observation (at least in mild cases), so the authors should explain this discrepancy.

4) In Figure 4H, I couldn't see any of the epithelial ligand expression in the plot. The authors should either provide enlarged pictures or immunostaining for validation.

5) The authors predicted several ligand/receptor interactions between cell types using NicheNet, and stated that these interactions may be crucial for the inflammation associated with the disease. The differential expression of these target genes could be simply a reflection of the inflammatory state rather than its cause. The authors need to provide some functional study showing these genes are actually important for the disease progression. Alternatively, a deeper discussion of this topic would be helpful.

6) The authors comment that; In our FACS gating strategy, neutrophils co-localised extensively with cell debris and were therefore excluded to avoid contamination. This raises a few points that need to be commented upon in the manuscript.

7) Neutrophils are not examined. This needs to be highlighted early in the manuscript as a non-expert would not necessarily know these are the most abundant immune population in health gingiva and that it would be expected to see them.

8) By not examining neutrophils the immune profiling of the manuscript is limited. This does not negate any of the important information shown in the manuscript but is a caveat when they authors suggest they are detailing the immune landscape. This needs to be acknowledged.

9) Why do the neutrophils co-localise with the cell debris? This raises an issue true of any single-cell examination of tissues – the populations that can be examined are dependent on the tissue digestion method/process; a different isolation protocol could generate slightly different results. This should be briefly commented upon.

10) It is unclear where the mild/onset and severe disease samples come from. In the clinical information provided, disease for the periodontitis patients was stated as being probing depth of ≥6mm – surely this would be "severe" disease? I understand that the catagories were based upon the authors own H and E but what was the initial tissue source. Same patient but different sites in their mouth?

11) For the data shown in 1E it would be nice to present an over-view of increases and decreases in populations across disease – a simple line graph showing proportions of cells in health, mild and severe disease which could go in the supplemental.

12) It is surprising that the two clusters of B cells cannot be further subsetted, similar to that which has been achieved for epithelial and mesenchymal cells. Is there substantial heterogeneity within IgG plasma cells and follicular B cells which prevents this?

13) Given the focus on T cells in the development and/or pathogenesis of periodontitis, an examination of the T cell subsets identified should be undertaken. Similar to what has been done for B cells in Figure 5.

---

## [Author Response]

Essential revisions:1) For the integration analysis, the authors should state how many cells were analyzed from each healthy and periodontitis patient.

Cells analysed were 4639 for “Healthy”, 4401 for “Mild” and 3367 for “Severe”. This has been added to the Figure 1 legend.

2) Using computational analysis, the authors identified a subpopulation of stromal cells expressing IGHBP4 and AEBP1 that are in a less differentiated state. It would be helpful to analyze some human gingival tissue and see where these AEBP1+ cells are located.

We analysed the location of AEBP1+ cells using immunohistochemistry, and observed that these cells are concentrated in the subepithelial area and at the tips of the dermal papilla regions. Based on their location, gene expression profile and similar observations in other studies (Kinchen et al., 2018), we suggest that these cells may play a role in tissue maintenance. We have added an additional panel in Figure 2 (G) showing the location of these cells.

3) In the integration analysis, it seems that epithelium is first affected relative to the stromal population in periodontitis patients. Even in the mild cases, the epithelial cell number is dramatically reduced. However, histology is not echoing this observation (at least in mild cases), so the authors should explain this discrepancy.

The proportion of cell numbers detected at each disease state are somewhat skewed by the total number of each population present in the whole sample. As an illustrative example, in the mild case we have a sharp increase in multiple immune cell subtypes, as the total numbers of these cells increase it would mean that their likelihood of being captured (when compared to other cell types) also increases as the 10x Chip will only capture a finite number of cells. Histology is not biased in this way as we can interrogate all cells in the tissue section, therefore to a small degree, the sequencing is biased to capturing cells that are in higher availability within a sample suspension than those that are not.

4) In Figure 4H, I couldn't see any of the epithelial ligand expression in the plot. The authors should either provide enlarged pictures or immunostaining for validation.

Figure 4 has been improved, including panel H showing epithelial ligand expression.

5) The authors predicted several ligand/receptor interactions between cell types using NicheNet, and stated that these interactions may be crucial for the inflammation associated with the disease. The differential expression of these target genes could be simply a reflection of the inflammatory state rather than its cause. The authors need to provide some functional study showing these genes are actually important for the disease progression. Alternatively, a deeper discussion of this topic would be helpful.

Due to the current availability of fresh samples under the existing COVID-enforced public health measures we have been unable to obtain sufficient fresh tissue to perform an informative functional analysis. However, *IL1B* and *TNF* have been documented in the literature as key components of the inflammatory cascade in periodontitis. We have discussed this in the revised manuscript, re-iterated in the results and the Discussion section that these predictions are highlighted as potentially beneficial intervention targets for future studies aiming to ameliorate the disease phenotype. We agree that distinguishing cause from effect is challenging, however relating single-cell data to clinical responses, cell-cell interactions (e.g. epithelial and stromal) can help inform disease aetiology and highlight new opportunities for clinical therapies.

6) The authors comment that; In our FACS gating strategy, neutrophils co-localised extensively with cell debris and were therefore excluded to avoid contamination. This raises a few points that need to be commented upon in the manuscript.7) Neutrophils are not examined. This needs to be highlighted early in the manuscript as a non-expert would not necessarily know these are the most abundant immune population in health gingiva and that it would be expected to see them.8) By not examining neutrophils the immune profiling of the manuscript is limited. This does not negate any of the important information shown in the manuscript but is a caveat when they authors suggest they are detailing the immune landscape. This needs to be acknowledged.9) Why do the neutrophils co-localise with the cell debris? This raises an issue true of any single-cell examination of tissues – the populations that can be examined are dependent on the tissue digestion method/process; a different isolation protocol could generate slightly different results. This should be briefly commented upon.

We agree with the reviewers’ comments in points 6 to 9 that this is a limitation of the study that needs to be highlighted. We also agree that neutrophil capturing/sequencing is heavily dependent on the digestion and isolation protocol used. Previous groups have reported on this limitation in similar studies (Smillie et al., 2019), and 10x Genomics openly acknowledge technical and biological limitations in capturing and efficiently capturing neutrophils and granulocytes (https://kb.10xgenomics.com/hc/en-us/articles/360004024032-Can-I-process-neutrophils-or-other-granulocytes-using-10x-Single-Cell-applications-).

Our digestion protocol (Whole Skin Dissociation Kit, human, Miltenyi, Bergisch-Gladbach, Germany) is optimised to digest fibrous tissue such as gingiva, and is possibly too harsh for a number of haematopoietic cells. Regarding neutrophils specifically, due to their small size neutrophils are notoriously difficult to pellet and subsequently capture, especially when intermingled with more bulky cell types, such as epithelium and mesenchymal lineages which will be preferentially captured due to their size and higher abundance. In addition to their small size, in absence of appropriate immunostaining, makes them particularly difficult to distinguish from larger cell debris in a FACS plot using forward and side scatter. Bioinformatically, due to their low abundance of RNA, neutrophils tend to be filtered out from subsequent analysis as it is unclear if these are low RNA expressing cells (naturally) or low sequencing quality cells.

We have made a suitable declaration in the Discussion and Materials and methods section of the manuscript.

10) It is unclear where the mild/onset and severe disease samples come from. In the clinical information provided, disease for the periodontitis patients was stated as being probing depth of ≥6mm – surely this would be "severe" disease? I understand that the catagories were based upon the authors own H and E but what was the initial tissue source. Same patient but different sites in their mouth?

Both diseased tissues were collected from two patients undergoing surgical periodontal treatment (pocket reduction surgeries) which is recommended when there is persistence of periodontal pockets following nonsurgical therapies. Thus, both patients were diagnosed with chronic periodontitis. Tissues were collected from LR6 buccal (mild) and LR7 buccal (severe) with probing depths of 6mm and 9mm respectively. Because we observed a distinct severity phenotype at a tissue and transcriptomic level, we staged the two samples as mild and severe.

11) For the data shown in 1E it would be nice to present an over-view of increases and decreases in populations across disease – a simple line graph showing proportions of cells in health, mild and severe disease which could go in the supplemental.

This has been added in Figure 1—figure supplement 3.

12) It is surprising that the two clusters of B cells cannot be further subsetted, similar to that which has been achieved for epithelial and mesenchymal cells. Is there substantial heterogeneity within IgG plasma cells and follicular B cells which prevents this?

There is not substantial heterogeneity within these patients to allow us to re-cluster these cells. An attempt was made, and we have included it as Figure 5—figure supplement 1, but this did not resolve the B-Cells substantially using UMAP projection. We have made appropriate comments in the Discussion and in the relevant result section to indicate this. We have excluded this supplemental figure from the main text as the data shown does not add nor impact the statements already made within the manuscript. For clarity we have included re-clustering of B and T cells in Figure 5—figure supplement 1.

13) Given the focus on T cells in the development and/or pathogenesis of periodontitis, an examination of the T cell subsets identified should be undertaken. Similar to what has been done for B cells in Figure 5.

In our analysis we can broadly identify CD4^+^ and CD8^+^ T-cells due to expression of the respective receptor genes. We could not identify discreet T-Cell subsets due to these largely being driven by VDJ genetic recombination.

Further re-clustering allowed us to broadly distinguish between CD4^+^ and CD8^+^ T cells, but did not provide extra information in transcriptionally distinct subsets present within the tissue. This could be a consequence of the isolation procedure being optimised for fibrous tissue. Some evidence pointing to this comes from T-cells having a low nFeature detection meaning that we could not capture as high a number of mRNA molecules from these cells as we did from others. Please see Figure 5—figure supplement 1. These limitations appear to be cell specific as difficulties in sequencing T-cells to a sufficient depth have been recently documented (Ding et al., 2020). This study showed that in order to classify these cells appropriately a high number of them has to be captured.